# An accurate and robust imputation method scImpute for single-cell RNA-seq data

Wei Vivian Li [1] & Jingyi Jessica Li [1,2]

The emerging single-cell RNA sequencing (scRNA-seq) technologies enable the investigation of transcriptomic landscapes at the single-cell resolution. ScRNA-seq data analysis is complicated by excess zero counts, the so-called dropouts due to low amounts of mRNA sequenced within individual cells. We introduce scImpute, a statistical method to accurately and robustly impute the dropouts in scRNA-seq data. scImpute automatically identifies likely dropouts, and only perform imputation on these values without introducing new biases to the rest data. scImpute also detects outlier cells and excludes them from imputation. Evaluation based on both simulated and real human and mouse scRNA-seq data suggests that scImpute is an effective tool to recover transcriptome dynamics masked by dropouts. scImpute is shown to identify likely dropouts, enhance the clustering of cell subpopulations, improve the accuracy of differential expression analysis, and aid the study of gene expression dynamics.

[1] Department of Statistics, University of California, Los Angeles, CA 90095-1554, USA. [2] Department of Human Genetics, University of California, Los Angeles, CA 90095-7088, USA. Correspondence and requests for materials should be addressed to J.J.L. (email: jli@stat.ucla.edu)

Bulk cell RNA-sequencing (RNA-seq) technology has been widely used for transcriptome profiling to study transcriptional structures, splicing patterns, and gene and transcript expression levels[1]. However, it is important to account for cell-specific transcriptome landscapes in order to address biological questions, such as the cell heterogeneity and the gene expression stochasticity[2]. Despite its popularity, bulk RNA-seq does not allow people to study cell-to-cell variation in terms of transcriptomic dynamics. In bulk RNA-seq, cellular heterogeneity cannot be addressed since signals of variably expressed genes would be averaged across cells. Fortunately, single-cell RNA sequencing (scRNA-seq) technologies are now emerging as a powerful tool to capture transcriptome-wide cell-to-cell variability[3–5]. ScRNA-seq enables the quantification of intra-population heterogeneity at a much higher resolution, potentially revealing dynamics in heterogeneous cell populations and complex tissues[6].

One important characteristic of scRNA-seq data is the "dropout" phenomenon where a gene is observed at a moderate expression level in one cell but undetected in another cell[7]. Usually, these events occur due to the low amounts of mRNA in individual cells, and thus a truly expressed transcript may not be detected during sequencing in some cells. This characteristic of scRNA-seq is shown to be protocol-dependent. The number of cells that can be analyzed with one chip is usually no more than a few hundreds on the Fluidigm C1 platform, with around 1–2 million reads per cell. On the other hand, protocols based on droplet microfluidics can parallelly profile >10,000 cells, but with only 100–200 k reads per cell[8]. Hence, there is usually a much higher dropout rate in scRNA-seq data generated by the droplet microfluidics than the Fluidigm C1 platform. New droplet-based protocols, such as inDrop[9] or 10x Genomics[10], have improved molecular detection rates but still have relatively low sensitivity compared to microfluidics technologies, without accounting for sequencing depths[11]. Statistical or computational methods developed for scRNA-seq need to take the dropout issue into consideration; otherwise, they may present varying efficacy when applied to data generated from different protocols.

Methods for analyzing scRNA-seq data have been developed from different perspectives, such as clustering, cell type identification, and dimension reduction. Some of these methods address the dropout events in scRNA-seq by implicit imputation while others do not. SNN-Cliq is a clustering method that uses scRNA-seq to identify cell types[12]. Instead of using conventional similarity measures, SNN-Cliq uses the ranking of cells/nodes to construct a graph from which clusters are identified. CIDR is the first clustering method that incorporates imputation of dropout values, but the imputed expression value of a particular gene in a cell changes each time when the cell is paired up with a different cell[13]. The pairwise distances between every two cells are later used for clustering. Seurat is a computational strategy for spatial reconstruction of cells from single-cell gene expression data[14]. It infers the spatial origins of individual cells from the cell expression profiles and a spatial reference map of landmark genes. It also includes an imputation step to impute the expression of landmark genes based on highly variable or so-called structured genes. ZIFA is a dimensionality reduction model specifically designed for zero-inflated single-cell gene expression analysis[15]. The model is built upon an empirical observation: dropout rate for a gene depends on its mean expression level in the population, and ZIFA accounts for dropout events in factor analysis.

Since most downstream analyses on scRNA-seq, such as differential gene expression analysis, identification of cell-type-specific genes, reconstruction of differentiation trajectory, and all the analyses mentioned earlier, rely on the accuracy of gene expression measurements, it is important to correct the false zero expression due to dropout events in scRNA-seq data by model-based imputation methods. To our knowledge, MAGIC is the first available method for explicit and genome-wide imputation of single-cell gene expression profiles[16]. MAGIC imputes missing expression values by sharing information across similar cells, based on the idea of heat diffusion. A key step in this method is to create a Markov transition matrix, constructed by normalizing the similarity matrix of single cells. In the imputation of a single cell, the weights of the other cells are determined by the transition matrix. During the preparation of this manuscript, we also noticed another imputation method SAVER[17], which borrows information across genes using a Bayesian approach to estimate (unobserved) true expression levels of genes. Both MAGIC and SAVER would alter all gene expression levels including those unaffected by dropouts, and this would potentially introduce new biases into the data and possibly eliminate meaningful biological variation. We think it is also inappropriate to treat all zero counts as missing values, since some of them may reflect true biological non-expression. Therefore, we propose a new imputation method for scRNA-seq data, scImpute, to simultaneously determine which values are affected by dropout events in data and perform imputation only on dropout entries. To achieve this goal, scImpute first learns each gene's dropout probability in each cell based on a mixture model. Next, scImpute imputes the (highly probable) dropout values in a cell by borrowing information of the same gene in other similar cells, which are selected based on the genes unlikely affected by dropout events (Fig. 1).

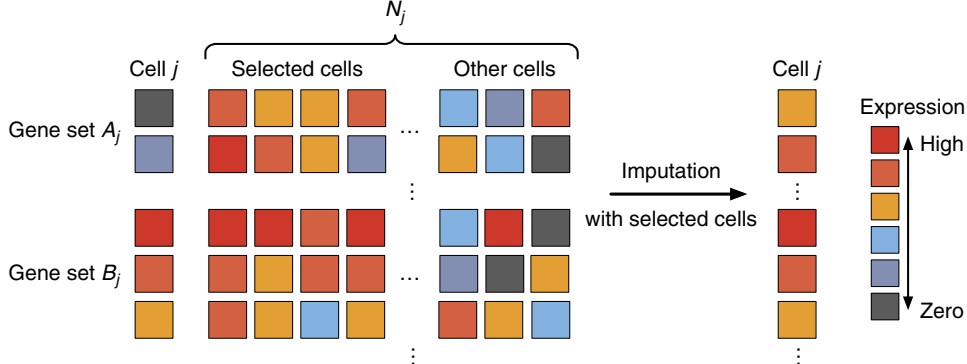

**Fig. 1** A toy example illustrating the workflow in the imputation step of scImpute method. scImpute first learns each gene's dropout probability in each cell by fitting a mixture model. Next, scImpute imputes the (highly probable) dropout values in cell $j$ (gene set $A_j$) by borrowing information of the same gene in other similar cells, which are selected based on gene set $B_j$ (not severely affected by dropout events). The details are described in Eqs. (2) and (3)

## Results

**scImpute recovers gene expression affected by dropouts.** A key reason for performing imputation on scRNA-seq data is to recover biologically meaningful transcriptome dynamics in single cells so that we can determine cell identity and identify differentially expressed (DE) genes among different cell types. We first use three examples to illustrate scImpute's efficacy in imputing gene expressions. (All the imputation results are obtained without using true cell type information unless otherwise noted.)

First, we show that scImpute recovers the true expression of the ERCC spike-in transcripts[18], especially low-abundance transcripts that are impacted by dropout events. The ERCC spike-ins are synthetic RNA molecules with known concentrations, which serve as gold standards of true expression levels, so that the read counts can be compared with the true expression for accuracy evaluation. The dataset contains 3005 cells from the mouse somatosensory cortex region[19]. After imputation, the median correlation (of the 3005 cells) between the 57 transcripts' read counts and their true concentrations increases from 0.92 to 0.95, and the minimum correlation increases from 0.81 to 0.89 (Supplementary Fig. 1). The read counts and true concentrations also present a stronger linear relationship in every single cell (Fig. 2).

Second, we show that scImpute correctly imputes the dropout values of 892 annotated cell cycle genes in 182 embryonic stem cells (ESCs) that had been staged for cell cycle phases (G1, G2M, and S)[20]. These genes are known to modulate the cell cycle and are expected to have non-zero expression during different stages of the cell cycle. Before imputation, 22.5% raw counts of the cell cycle genes are zeros, which are highly likely due to dropouts. The data are normalized by sequencing depths instead of ERCC spike-ins as described in McDavid et al.[21]. After imputation, most of the dropout values are corrected, and true dynamics of these genes in the cell cycle are revealed (Supplementary Figs. 2 and 3). The imputed counts also better represent the true biological variation in these cell cycle genes (Fig. 3).

Third, we use a simulation study to illustrate the efficacy of scImpute in enhancing the identification of cell types. We simulate expression data of three cell types $c_1$, $c_2$, and $c_3$, each with 50 cells, and 810 among 20,000 genes are truly differentially expressed (DE) (details in the Methods section). Even though the three cell types are clearly distinguishable when we apply principal component analysis (PCA) to the complete data, they become less well separated in the raw data with dropout events. The within-cluster sum-of-squares calculated based on the first two principal components (PCs) increases from 94 in the complete data to 2646 in the raw data. However, the relationships among the 150 cells are clarified after we apply scImpute. The

other two methods MAGIC and SAVER are also able to distinguish the three cell types, but MAGIC introduces artificial signals that largely alter the data and thus the PCA result, while SAVER only slightly improves the clustering result over that of the raw data (Fig. 4). In addition, the dropout events obscure the differential pattern and thus increase the difficulty of detecting DE genes. The imputed data by scImpute lead to a clearer contrast between the upregulated genes in different cell types, while the imputed data by MAGIC and SAVER fail to recover this pattern (Fig. 4). We also assess how the prevalence of dropout values influences the performance of scImpute. As expected, the DE analysis based on the imputed data has increased accuracy as the dropout proportion decreases. Yet scImpute still achieves > 80.0% area under the precision-recall curve even when the proportion of zero counts in the raw data is as high as 75.0% (Supplementary Fig. 4).

**scImpute improves the identification of cell subpopulations.** To illustrate scImpute's capacity in aiding the identification of cell types or cell subpopulations, we apply our method to two real scRNA-seq datasets. The first one is a smaller dataset of mouse preimplantation embryos[22]. It contains RNA-seq profiles of 268 single cells from 10 developmental stages. Partly due to dropout events, 70.0% of read counts in the raw count matrix are zeros. To illustrate the dropout phenomenon, we plot the $\log_{10}$-transformed read counts of two 16-cell stage cells as an example in Supplementary Fig. 5. Even though the two cells come from the same stage, many expressed genes have zero counts in only one cell. This problem is alleviated in the imputed data by scImpute, and the Pearson correlation between the two cells increases from 0.72 to 0.82 (Supplementary Fig. 5), primarily due to the decreased number of genes only expressed in one cell. MAGIC achieves an even higher correlation (0.95) but also introduces excess large counts that do not exist in the raw data. Biological variation between the two cells is likely lost in the imputation process of MAGIC. On the other hand, SAVER's imputation does not have a clear impact on the data.

We compare the imputation results by investigating the clustering accuracy in the first two PCs. Although it is possible to differentiate the major developmental stages from the raw data, the imputed data by scImpute output more compact clusters (Fig. 5). MAGIC gives a clean pattern of developmental stages, but it has a high risk of removing biologically meaningful variation, given that many cells of the same stage have almost identical scores in the first two PCs. scImpute is the only method that is able to detect outlier cells. We then compare the clustering results of the spectral clustering algorithm[23] on the first two PCs. Since the true cluster labels include several sub-stages in

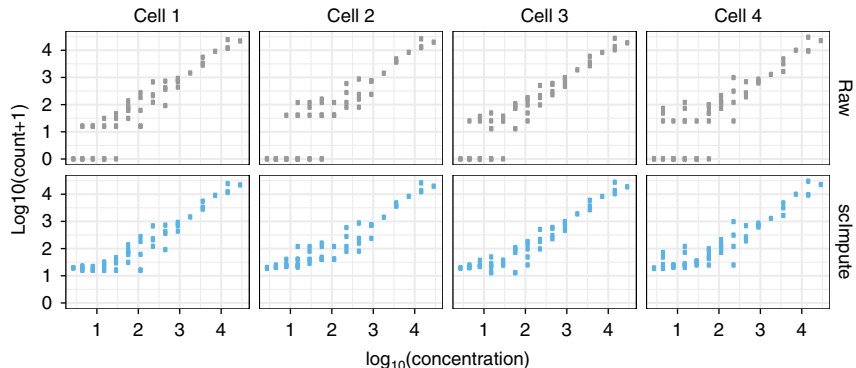

**Fig. 2** scImpute improves the dropouts in the ERCC RNA transcripts. The y-axis and x-axis give the ERCC spike-ins' $\log_{10}$(count+1) and $\log_{10}$(concentration) in four randomly selected mouse cortex cells. The imputed data present stronger linear relationships between the true concentrations and the observed counts

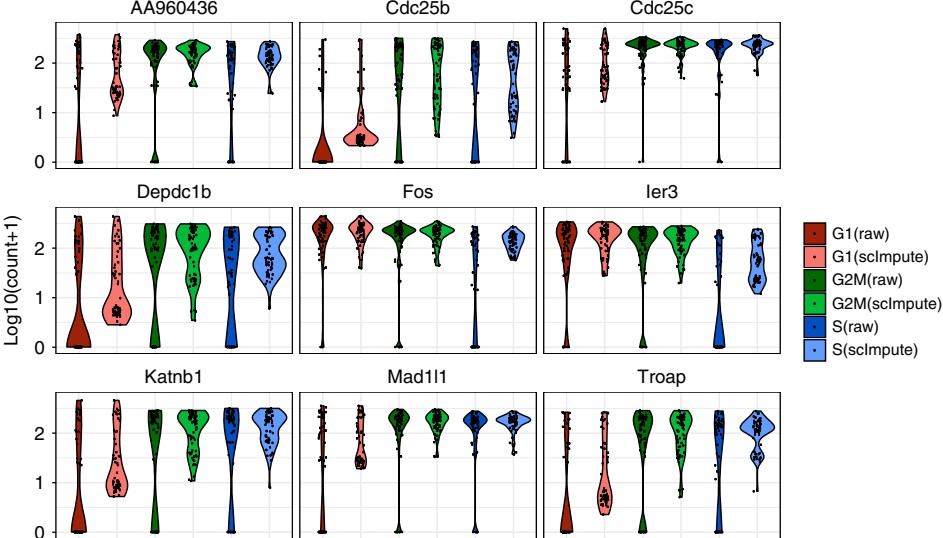

**Fig. 3** Violin plots showing the $\log_{10}$(count+1) of nine cell cycle genes. The expression levels of these genes belong to three phases (G1, G2M, and S). scImpute has corrected the dropout values of cell cycle genes

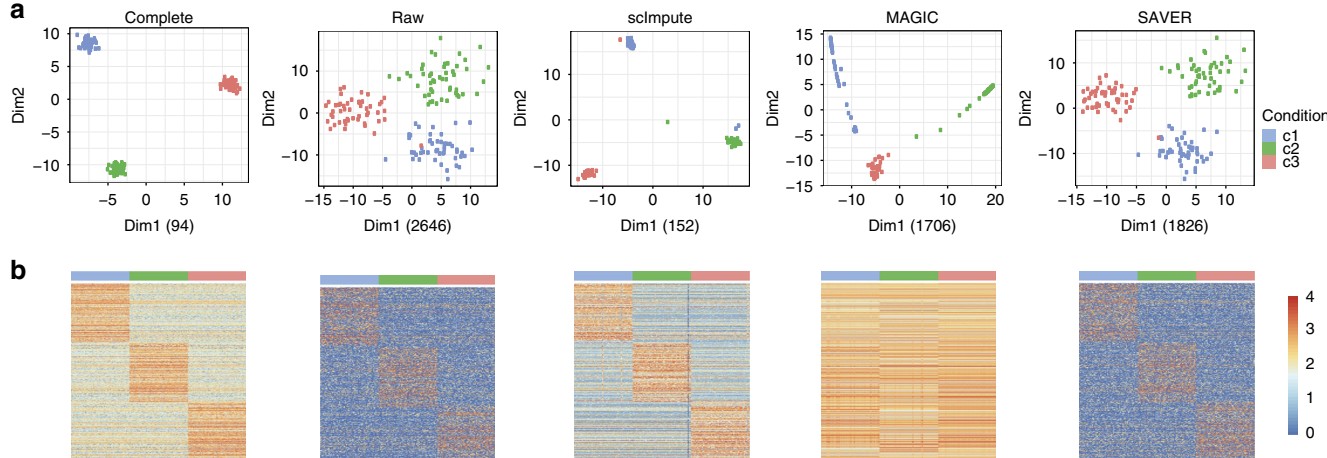

**Fig. 4** scImpute corrects dropout values and helps define cellular identity in the simulated data. **a** The first two PCs calculated from the complete data, the raw data, and the imputed data by scImpute, MAGIC, and SAVER. Numbers in the parentheses are the within-cluster sum of squares calculated based on the first two PCs. The within-cluster sum of squares is defined as $\sum_{k=1}^{3} \sum_{j=1}^{50} \left\| \mathbf{y}_{kj} - \overline{\mathbf{y}}_{k\cdot} \right\|^2$, where $\overline{\mathbf{y}}_{k\cdot} = \frac{1}{50}\sum_{j=1}^{50} \mathbf{y}_{kj}$ and $\mathbf{y}_{kj}$ is a vector of length 2, denoting the first two PCs of cell $j$ in cell type $c_k$. **b** The expression profiles of the 810 true DE genes in the complete, raw, and imputed datasets

embryonic development, we use different numbers of clusters, $k$ = 6, 8, 10, 12, and 14. The results are evaluated by four different measures: adjusted Rand index[24], Jaccard index[25], normalized mutual information (nmi)[26], and purity (Methods section). The four measures are all between 0 and 1, with 1 indicating a perfect match between the clustering result and the truth. All the four measures indicate that scImpute leads to the best clustering result as compared with no imputation and the imputation by MAGIC or SAVER (Supplementary Fig. 6). This result suggests that scImpute improves the clustering of cell subpopulations by imputing dropout values in scRNA-seq data.

We also apply scImpute to a large dataset generated by the high-throughput droplet-based system[10]. The dataset contains 4500 peripheral blood mononuclear cells (PBMCs) of nine immune cell types, with 500 cells of each type. In the raw data, 92.6% read counts are exactly zeros. Given dimension reduction by t-SNE[27], the cytotoxic and naive cytotoxic T cells are clustered together, and the other four types of T cells are not separated. After scImpute's imputation, the cytotoxic (label 11) and naive

cytotoxic T cells (label 8) are separated into two groups, and the naive T cells (label 5) and memory T cells (label 3) are now distinguishable from the remaining T cells (Fig. 6). This evidence shows the strong ability of scImpute to identify cell subpopulations despite missing cell type information. On the other hand, MAGIC does not improve the clustering of cells in the same type (Supplementary Fig. 7), and we could not obtain SAVER's results after running the program overnight. After the imputation by scImpute, the monocyte cells are grouped into one large and two small clusters, and we find that the three clusters reveal dynamics of two signature genes, *FCER1A*, which accumulates during the dendritic cell differentiation from monocytes[28], and *S100A8*, whose expression differs among subsets of human monocytes[29] (Fig. 6 and Supplementary Fig. 8). The large cluster (label 10) is characterized by high expression of *S100A8* and moderate expression of *FCER1A*; one of the small clusters (label 1) presents high expression levels of both *S100A8* and *FCER1A*, while in the other small cluster (label 2) *FCER1A* is mostly non-expressed. We also investigate the three clusters (labels 6, 9, and 12) of

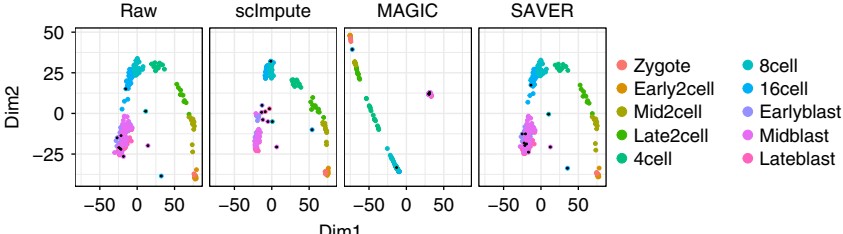

**Fig. 5** scImpute improves cell subpopulation clustering in the mouse embryonic cells. The scatter plots show the first two PCs obtained from the raw and imputed data of mouse embryonic cells. The black dots mark the outlier cells detected by scImpute

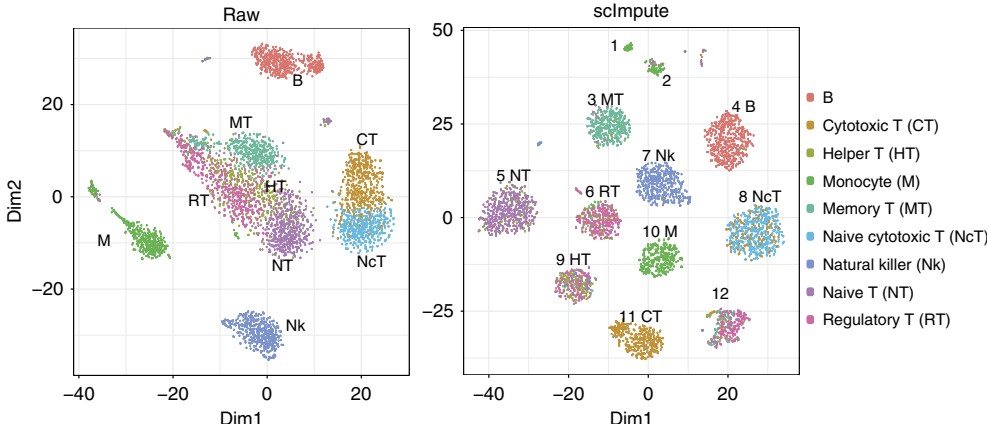

**Fig. 6** scImpute helps identify cell subpopulations in the PBMC dataset. The scatter plots give the first two dimensions of the t-SNE results calculated from raw and imputed PBMC dataset. Numbers marked on the imputed data are cluster labels. Cell type information is marked for major clusters. We note that for the raw data, we did not mask zero expression values as missing values in the dimension reduction and the clustering steps

regulatory/memory/helper T cells. The three clusters are supported by the expression of eight potential marker genes (*ACTG1*, *ATP5C1*, *CCT8*, *CIRBP*, *DUSP1*, *FLNA*, *FOS*, and *GAPDH*): cells in the same cluster have a similar expression pattern (Supplementary Fig. 9). This example shows that scImpute provides an opportunity to discover new cell sub-populations and their marker genes.

**scImpute assists differential gene expression analysis**. ScRNA-seq data provide insights into the stochastic nature of gene expression in single cells but suffer from a relatively low signal-to-noise ratio compared with bulk RNA-seq data. Thus an effective imputation method should lead to a better agreement between scRNA-seq and bulk RNA-seq data of the same biological condition on genes known to have little cell-to-cell heterogeneity. To evaluate whether the DE genes identified from single-cell data are more accurate after imputation, we utilize a real dataset with both bulk and single-cell RNA-seq experiments on human embryonic stem cells (ESC) and definitive endoderm cells (DEC)[30]. This dataset includes six samples of bulk RNA-seq (four in H1 ESC and two in DEC) and 350 samples of scRNA-seq (212 in H1 ESC and 138 in DEC). The percentages of zero gene expression are 14.8% in bulk data and 49.1% in single-cell data.

We apply scImpute, MAGIC, and SAVER to impute the gene expression for each cell type respectively, and then perform DE analysis on the raw data and the imputed data by each method, respectively. We use the R package DESeq2[31] to identify DE genes from the bulk data, and the R packages DESeq2 and MAST[32] to identify DE genes from the scRNA-seq data. Inspecting the top 200 DE genes from the bulk data, we find that their expression profiles in the scRNA-seq data have stronger concordance with those in the bulk data after imputation by scImpute

(Supplementary Fig. 10). We apply different thresholds to false discovery rates (FDRs) of genes in the bulk data to obtain a DE gene list for every threshold. The same thresholds are applied to the FDRs of genes calculated from the raw and imputed scRNA-seq datasets to obtain DE gene lists respectively. Then we compare the DE gene lists obtained from the scRNA-seq data with those from the bulk data (i.e., the standard) to calculate precision and recall rates and *F*-scores (Supplementary Fig. 11). scImpute leads to more similar DE gene lists to those from the bulk data and achieves around 10% higher *F*-scores compared with results on the raw data. We find that scImpute makes a right balance between the precision and recall rate, while MAGIC has low precision, and SAVER has low-recall rate and is barely distinguishable from no imputation. We conclude that scImpute is preferred when users have a priority on the precision of the DE genes.

A comparison between the expression profiles of DEC and ESC marker genes[30, 33, 34] shows that the imputed data by scImpute best reflect the gene expression signatures by removing undesirable technical variation resulted from dropouts (Fig. 7a and Supplementary Fig. 13). To determine whether the DE genes identified in scRNA-seq data are biologically meaningful, we performed gene ontology (GO) enrichment analysis[35]. In the ~300 DEC upregulated genes that are only detected in the imputed data by scImpute but not in the raw data, enriched GO terms are highly relevant to the functions of DECs (Fig. 7c, Supplementary Fig. 12 and Supplementary Tables 4 and 5). However, in the ~300 DEC upregulated genes that are only detected in the raw data, enriched GO terms are general and not characteristic to DECs (Supplementary Tables 6 and 7). These results also demonstrate that scImpute can facilitate the usage of DE methods that were not designed for single-cell data.

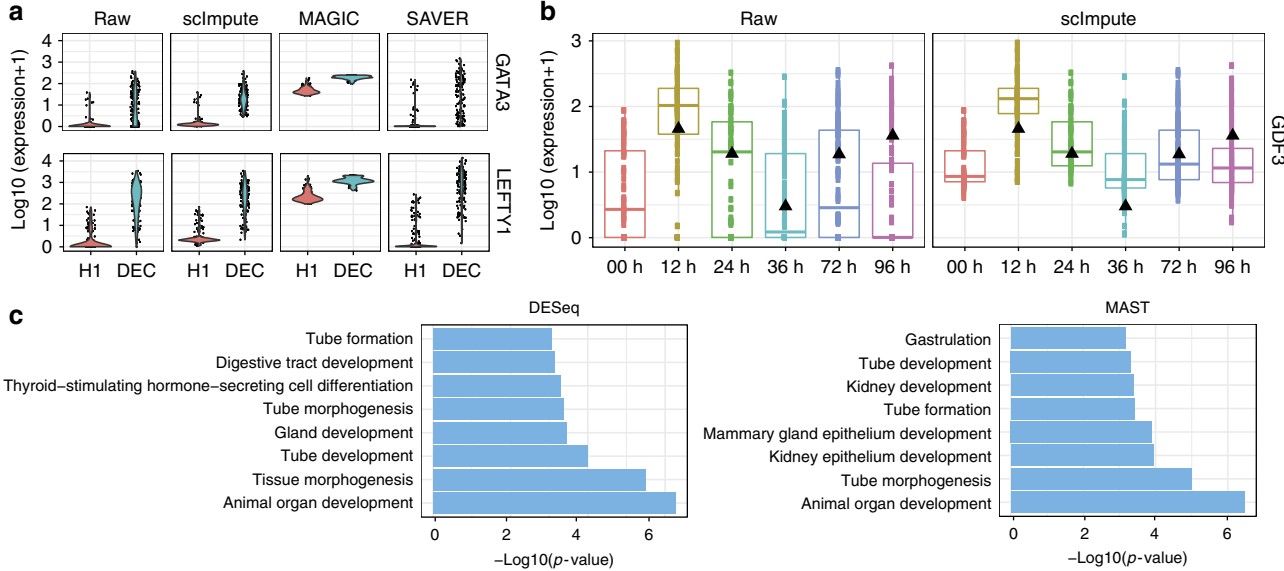

**Fig. 7** scImpute improves differential gene expression analysis and reveals expression dynamics in time-course experiments. **a** Raw and imputed expression levels of two marker genes of DEC. **b** Time-course expression patterns of the gene *GDF3*, which is annotated with the GO term "endoderm development." Black triangles mark the gene's expression in bulk data. **c** Selected GO terms enriched in the DEC-upregulated genes that can be only detected (by DESeq2 or MAST) in the imputed data by scImpute, but not in the raw data

**scImpute recovers gene expression temporal dynamics**. Aside from the data we used in differential expression analysis, Chu et al.[30] also generated bulk and single-cell time-course RNA-seq data profiled at 0, 12, 24, 36, 72, and 96 h of differentiation during DEC emergence (Supplementary Table 2). We utilize this dataset to show that scImpute can help recover the DE signals that are difficult to identify in the raw time-course data, and reduce false discoveries resulted from dropouts. We first apply scImpute to the raw scRNA-seq data with true cell type labels, and then study how the time-course expression patterns change in the imputed data. The imputed data better distinguish cells of different time points (Supplementary Fig. 14), suggesting that imputed read counts reflect more accurate transcriptome dynamics along the time course. Even though the scRNA-seq data present more biological variation than the bulk data, it is reasonable to expect that the average gene expression signal across cells in scRNA-seq should correlate with the signal in bulk RNA-seq. For a genome-wide comparison, the imputed data have significantly higher Pearson correlations with the bulk data (Supplementary Fig. 15). We study 70 genes associated with the GO term "endoderm development"[36] and found that a subset of these genes that are likely affected by dropout events show higher expression and better consistency with the bulk data after the imputation by scImpute (Fig. 7b and Supplementary Fig. 16). Similarly, we also study the marker genes (e.g., *FOXA2, HHEX*, and *CXCR4*) of DECs[30, 33, 34] and these genes' expression levels at time point 96 h are recovered by scImpute even though they have a median read count of zero in the raw data (Supplementary Fig. 17).

## Discussion

We propose a statistical method scImpute to address the dropout events prevalent in scRNA-seq data. scImpute focuses on imputing the missing expression values of dropout genes, while retaining the expression levels of genes that are largely unaffected by dropout events. Hence, scImpute can reduce technical variation resulted from scRNA-seq and better represent cell-to-cell biological variation, while it also avoids introducing excess biases during its imputation process. To achieve the above goals, scImpute first learns each gene's dropout probability in each cell

by fitting a mixture model for each cell type. Next, scImpute imputes the (highly probable) dropout values of genes in a cell by borrowing information of the same gene in other similar cells, which are selected based on the genes not severely affected by dropout events. Comprehensive studies on both simulated and real data suggest that compared with the raw scRNA-seq data, the imputed data by scImpute better present cell type identity and lead to more accurate DE analysis results.

An attractive advantage of scImpute is that it can be incorporated into most existing pipelines or downstream analysis of scRNA-seq data, such as normalization[4, 37], differential expression analysis[7, 32], clustering and classification[12, 13], etc. Despite the availability of computational methods that directly model zero-inflation in data[7, 32], scImpute takes the imputation perspective to improve the data quality, and its applicability is not restricted to a specific task. scImpute inputs the raw read count matrix and outputs an imputed count matrix of the same dimensions, so that it can be seamlessly combined with other computational tools without data reformatting or transformation. We also note that new analyzing tools specifically designed for the imputed scRNA-seq data by scImpute may have improved performance over existing methods developed for raw scRNA-seq data, by incorporating features, such as smaller proportions of zero expression, dropout rates, and dropout probabilities estimated by the mixture models. Another important feature of scImpute is that it only involves two parameters that can be easily understood and selected. The first parameter $K$ denotes the potential number of cell populations. It can be chosen based on the clustering result of the raw data and the resolution level desired by the users. If users are only interested in the differences among the major clusters, they could use a relatively small $K$, and scImpute can borrow more information among individual cells; otherwise, users can select a relatively large $K$, and scImpute would be more conservative in the imputation process. The second parameter is a threshold $t$ on dropout probabilities. We show in a sensitivity analysis that scImpute is robust to the different parameter values (Supplementary Fig. 20), and a default threshold value 0.5 is sufficient for most scRNA-seq data. Moreover, cell type information is not necessary for the scImpute method. When

cell type information is available, separate imputation for each cell type is expected to produce more accurate results. But as illustrated by simulation and real data results, scImpute is able to infer cell-type-specific expression even when the true cell type labels are not supplied.

scImpute scales up well when the number of cells increases, and the computation efficiency can be largely improved if a filtering step on cells can be performed based on biological knowledge. Aside from computational complexity, another future direction is to improve imputation efficiency further when dropout rates in the raw data are severely high, as with the droplet-based technologies. Imputation task becomes more difficult with a more substantial proportion of missing values. More complicated models that account for gene similarities may yield more accurate imputation results, but the prevalence of dropout events may require additional prior knowledge on similar genes to assist modeling.

## Methods

**Data processing and normalization.** The input of our method is a count matrix $X^C$ with rows representing genes and columns representing cells, and our eventual goal is to construct an imputed count matrix with the same dimensions. We start by normalizing the count matrix by the library size of each sample (cell) so that all samples have one million reads. Denoting the normalized matrix by $X^N$, we then make a matrix $X$ by taking the $\log_{10}$ transformation with a pseudo count 1.01:

$$X_{ij} = \log_{10}\left(X_{ij}^N + 1.01\right); \; i = 1, 2, \ldots, I, \; j = 1, 2, \ldots, J,$$

where $I$ is the total number of genes and $J$ is the total number of cells. The pseudo count is added to avoid infinite values in parameter estimation in a later step. The advantage of the logarithmic transformation is to prevent a few large observations from being extremely influential, and the transformed values become continuous, allowing for greater flexibility for the modeling.

**Detection of cell subpopulations and outliers.** Since scImpute borrows information of the same gene from similar cells to impute the dropout values, a critical step is to determine first which cells are from the same subpopulation. Due to excess zero counts in scRNA-seq data, it is difficult to cluster cells into true cell types accurately. Hence, the goal of this step is to find a candidate pool of "neighbors" for each cell. scImpute will select similar cells from the candidate neighbors in a subsequent imputation step. Suppose that scImpute clusters the cells in a dataset into $K$ subpopulations in this step. For each cell, its candidate neighbors are the other cells in the same cluster.

1. PCA is performed on matrix $X$ for dimension reduction and the resulting matrix is denoted as $Z$, where columns represent cells and rows represent principal components (PCs). The purpose of dimension reduction is to reduce the impact of large portions of dropout values. The PCs are selected such that at least 40% of the variance in data could be explained.

2. Based on the PCA-transformed data $Z$, the distance matrix $D_{J \times J}$ between the cells could be calculated. For each cell $j$, we denote its distance to the nearest neighbor as $l_j$. For the set $L = \{l_1, \ldots, l_J\}$, we denote its first quartile as $Q_1$, and third quartile as $Q_3$. The outlier cells are those cells which do not have close neighbors:

$$O = \left\{ j : l_j > Q_3 + 1.5(Q_3 - Q_1) \right\}.$$

For each outlier cell, we set its candidate neighbor set $N_j = \varnothing$. Please note that the outlier cells could be a result of experimental/technical errors or biases, but they may also represent real biological variation as rare cell types. scImpute would not impute gene expression values in outlier cells, nor use them to impute gene expression values in other cells.

3. The remaining cells $\{1, \ldots, J\} \backslash O$ are clustered into $K$ groups by spectral clustering[23]. We denote $g_j = k$ if cell $j$ is assigned to cluster $k$ ($k = 1, \ldots, K$). Hence, cell $j$ has the candidate neighbor set $N_j = \{j' : g_{j'} = g_j, j' \neq j\}$.

**Identification of dropout values.** Once we obtain the transformed gene expression matrix $X$ and the candidate neighbors of each cell $N_j$, the next step is to infer which genes are affected by the dropout events in which cells. Instead of treating all zero values as dropout events, we construct a statistical model to systematically determine whether a zero value comes from a dropout event or not. With the existence of dropout events, most genes have a bimodal expression pattern across similar cells, and that pattern can be described by a mixture model of two components (Supplementary Fig. 18). The first component is a Gamma distribution used to account for the dropouts, while the second component is a Normal

distribution to represent the actual gene expression levels. Please note that the transformed gene expression levels are no longer integers, so the widely used negative binomial distribution for read counts is not a proper choice here. For each gene, the proportions and parameters of the two components could be different in various cell types, so we construct separate mixture models for different cell subpopulations.

For each gene $i$, its expression in cell subpopulation $k$ is modeled as a random variable $X_i^{(k)}$ with density function

$$f_{X_i^{(k)}}(x) = \lambda_i^{(k)} \text{Gamma}\left(x; \alpha_i^{(k)}, \beta_i^{(k)}\right) + \left(1 - \lambda_i^{(k)}\right) \text{Normal}\left(x; \mu_i^{(k)}, \sigma_i^{(k)}\right), \quad (1)$$

where $\lambda_i^{(k)}$ is gene $i$'s dropout rate in cell subpopulation $k$, $\alpha_i^{(k)}, \beta_i^{(k)}$ are the shape and rate parameters of Gamma distribution, and $\mu_i^{(k)}, \sigma_i^{(k)}$ are the mean and standard deviation of Normal distribution. The intuition behind this mixture model is that if a gene has high expression and low variation in most cells within a cell subpopulation, a zero count is more likely to be a dropout value; on the other hand, if a gene has constantly low or medium expression with high variation, then a zero count may reflect real biological variability. An advantage of this model is that it does not assume an empirical relationship between dropout rates and mean gene expression levels, as Kharchenko et al.[7] did, thus allowing more flexibility in the model estimation. The parameters in the mixture model are estimated by the Expectation–Maximization (EM) algorithm, and we denote their estimates as $\hat{\lambda}_i^{(k)}$, $\hat{\alpha}_i^{(k)}$, $\hat{\beta}_i^{(k)}$, $\hat{\mu}_i^{(k)}$, and $\hat{\sigma}_i^{(k)}$. It follows that the dropout probability of gene $i$ in cell $j$, which belongs to subpopulation $k$, can be estimated as

$$d_{ij} = \frac{\hat{\lambda}_i^{(k)} \text{Gamma}\left(X_{ij}; \hat{\alpha}_i^{(k)}, \hat{\beta}_i^{(k)}\right)}{\hat{\lambda}_i^{(k)} \text{Gamma}\left(X_{ij}; \hat{\alpha}_i^{(k)}, \hat{\beta}_i^{(k)}\right) + \left(1 - \hat{\lambda}_i^{(k)}\right) \text{Normal}\left(X_{ij}; \hat{\mu}_i^{(k)}, \hat{\sigma}_i^{(k)}\right)}.$$

Therefore, each gene $i$ has an overall dropout rate $\hat{\lambda}_i^{(k)}$ in cell subpopulation $k$, which does not depend on individual cells within the subpopulation. Gene $i$ also has dropout probabilities $d_{ij}$ ($j = 1, 2, \ldots, J$), which may vary among different cells. During the preparation of this manuscript, it came to our attention that Ghazanfar et al.[38] also used the Gamma-Normal mixture model to analyze scRNA-seq data but only applied it to categorize non-zero expression values into low-expression values and high-expression values.

**Imputation of dropout values.** Now, we impute the gene expressions cell by cell. For each cell $j$, we select a gene set $A_j$ in need of imputation based on the genes' dropout probabilities in cell $j$: $A_j = \{i : d_{ij} \geq t\}$, where $t$ is a threshold on dropout probabilities. We also have a gene set $B_j = \{i : d_{ij} < t\}$ that have accurate gene expression with high confidence and do not need imputation. We learn cells' similarities through the gene set $B_j$. Then we impute the expression of genes in the set $A_j$ by borrowing information from the same gene's expression in other similar cells learned from $B_j$. Supplementary Figs. 19 and 20c give some real data distributions of genes' zero count proportions across cells and genes' dropout probabilities, showing that it is reasonable to divide genes into two sets. To learn the cells similar to cell $j$ from $B_j$, we use the non-negative least squares (NNLS) regression:

$$\widehat{\boldsymbol{\beta}}^{(j)} = \text{argmin}_{\boldsymbol{\beta}^{(j)}} \left\| X_{B_j, j} - X_{B_j, N_j} \boldsymbol{\beta}^{(j)} \right\|_2^2, \text{subject to } \boldsymbol{\beta}^{(j)} \geq \mathbf{0}. \quad (2)$$

Recall that $N_j$ represents the indices of cells that are candidate neighbors of cell $j$. The response $X_{B_j, j}$ is a vector representing the $B_j$ rows in the $j$-th column of $X$, the design matrix $X_{B_j, N_j}$ is a sub-matrix of $X$ with dimensions $|B_j| \times |N_j|$, and the coefficients $\boldsymbol{\beta}^{(j)}$ is a vector of length $|N_j|$. Note that NNLS itself has the property of leading to a sparse estimate $\widehat{\boldsymbol{\beta}}^{(j)}$, whose components may have exact zeros[39], so NNLS can be used to select similar cells of cell $j$ from its neighbors $N_j$. Finally, the estimated coefficients $\widehat{\boldsymbol{\beta}}^{(j)}$ from the set $B_j$ are used to impute the expression of genes in the set $A_j$ in cell $j$:

$$\hat{X}_{ij} = \begin{cases} X_{ij}, & i \in B_j, \\ X_{i, N_j} \widehat{\boldsymbol{\beta}}^{(j)}, & i \in A_j. \end{cases} \quad (3)$$

We construct a separate regression model for each cell to impute the expression of genes with high dropout probabilities (Fig. 1). scImpute simultaneously determines the values that need imputation, and would not introduce biases to the high expression values of accurately measured genes. Supplementary Fig. 21 illustrates the expression distributions of four example genes (before and after imputation) across the 268 cells in the mouse embryo dataset. Since scImpute corrects the identified dropouts, the proportion of zero expression is reduced in the imputed data. However, we can clearly see that scImpute does not inflate all the zero expressions, and some genes remain to have bimodal distributions after the imputation. Therefore, scImpute takes a relatively conservative approach to impute dropouts, attempting to avoid introducing biases and retain the stochasticity of gene expression.

The application of scImpute involves two parameters. The first parameter is $K$, which determines the number of initial clusters to help identify candidate

neighbors of each cell. The imputation results do not heavily rely on the choice of $K$ because scImpute uses a model-based method to select similar cells at a later stage. However, setting $K$ to a value close to the true number of cell subpopulations can assist the selection of similar cells. The second parameter is a threshold $t$, and the imputation is only applied to the genes with dropout probabilities larger than $t$ in a cell to avoid over-imputation. Please note that the threshold is set on the dropout probability (the probability that a gene being a dropout in a cell), not on the dropout rate (the proportion of cells in which the gene is affected by dropout events). The sensitivity analysis based on the mouse embryo data[22] suggests that scImpute is robust to varying parameter values (Supplementary Fig. 20a,b). Notably, the choice of parameter $t$ only affects a minute fraction of genes (Supplementary Fig. 20c).

**Generation of simulated scRNA-seq data**. We suppose there are three cell types $c_1$, $c_2$, and $c_3$, each with 50 cells, and there are 20,000 genes in total. In the gene population, only 810 genes are truly differentially expressed, with one third having higher expression in each cell type, respectively. We directly generate genes' $\log_{10}$-transformed read counts as expression values. First, mean expression levels of the 20,000 genes are randomly drawn from a Normal distribution with mean 1.8 and standard deviation 0.5. Standard deviations of the gene expression of the 20,000 genes are randomly drawn from a Normal distribution with mean 0.6 and standard deviation 0.1. These parameters are estimated from the real dataset of mouse embryo cells. Second, we randomly draw 270 genes and multiply each of their mean expression in cell type $c_1$ by an integer randomly sampled from $\{2, 3, \ldots, 10\}$; we also create 270 highly expressed genes for each of cell types $c_2$ and $c_3$ in the same way. Finally, the expression values of each gene in the 150 cells are simulated from Normal distributions defined by the mean and standard deviation parameters obtained in the first step, and shifted as described in the second step. We refer to the resulting gene expression data as the complete data. Finally, we suppose the dropout rate of each gene follows a double exponential function $\exp(-0.1 \times$ mean expression$^2)$, as assumed in[15]. Zero values are then introduced into the simulated data for each gene based on a Bernoulli distribution defined by the dropout rate of the gene, resulting in a gene expression matrix with excess zeros and in need of imputation. We refer to the gene expression data, which have zero values introduced, as the raw data. Please note that the generation of gene expression values does not directly follow the mixture model used in scImpute, so that we use this simulation to investigate the efficacy and robustness of scImpute in a fair way.

**Four evaluation measures of clustering results**. The four measures (adjusted Rand index, Jaccard index, normalized mutual information (nmi), and purity) focus on different properties of the clustering results. The adjusted Rand index penalizes both false positive and false negative decisions, where a positive decision means that two cells are clustered into one cluster, while a negative decision means that two cells are clustered into different clusters. The Jaccard index is similar to the adjusted Rand index, but it does not account for true negatives. The nmi measures the similarity from the perspective of information theory. The purity score is simply the percentage of the total number of samples that are from the same true class and clustered together correctly, and it does not penalize on splitting a true class into multiple clusters.

We choose $a$ to represent the number of observation pairs which are correctly grouped into the same class by the clustering method. $b$ represents the number of observation pairs which are grouped into the same cluster but actually belong to different classes. $c$ represents the number of observation pairs which are grouped into different clusters but actually belong to the same class. $d$ represents the number of observation pairs which are correctly grouped into different clusters (Supplementary Table 1).

We use $U = \{u_1, \ldots, u_P\}$ to denote the true partition of $P$ classes and $V = \{v_1, \ldots, v_K\}$ to denote the partition given by spectral clustering results. Let $n_i$ and $n_j$ be the numbers of observations in class $u_i$ and cluster $v_j$ respectively, and $n_{ij}$ denotes the number of observations in both class $u_i$ and cluster $v_j$.

The adjusted Rand index is calculated as

$$\frac{\sum_{i=1}^{P}\sum_{j=1}^{K}\binom{n_{ij}}{2} - \left[\sum_{i=1}^{P}\binom{n_i}{2}\sum_{j=1}^{K}\binom{n_j}{2}\right]/\binom{n}{2}}{\frac{1}{2}\left[\sum_{i=1}^{P}\binom{n_i}{2} + \sum_{j=1}^{K}\binom{n_j}{2}\right] - \left[\sum_{i=1}^{P}\binom{n_i}{2}\sum_{j=1}^{K}\binom{n_j}{2}\right]/\binom{n}{2}}$$

where $n = \sum_{i=1}^{P} n_i = \sum_{j=1}^{K} n_{.j}$.

The Jaccard index is calculated as

$$\frac{a}{a+b+c}.$$

The normalized mutual information is calculated as

$$\frac{2I(U,V)}{H(U)+H(V)},$$

where $I(U,V)$ is mutual information, and $H(U)$ and $H(V)$ are the entropies of partitions $U$ and $V$.

The purity score is calculated as

$$\frac{1}{n}\sum_{i}\max_{j}\left|v_i \cap u_j\right|.$$

**Data availability**. The scRNA-seq data used in this manuscript are all publicly available, and their sources are summarized in Supplementary Table 3. The ERCC spike-ins data are available at the Gene Expression Omnibus (GEO) under accession code GSE60361. The cell cycle data are available at ArrayExpress under accession code E-MTAB-2512. The mouse embryo data are available at GEO under accession code GSE45719. The PBMC dataset is available at 10x Genomics's official website (https://support.10xgenomics.com/single-cell-gene-expression/datasets). The human ESC and DEC data are available at GEO under accession code GSE75748. The R package scImpute is freely available at https://github.com/Vivianstats/scImpute.

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

## Acknowledgements

We are grateful to Douglas Arneson, Feiyang Ma, Dr. Robert Modlin, Dr. Matteo Pellegrini, and Dr. Xia Yang at University of California, Los Angeles, for providing insightful discussions. We thank Dr. Mark Biggin at Lawrence Berkeley National Laboratory for his suggestions on this manuscript. We also thank Dr. Daria Merkurjev for assisting the data collection. This work was supported by the PhRMA Foundation Research Starter Grant in Informatics, NIH/NIGMS grant R01GM120507, and NSF grant DMS-1613338.

## Author contributions

W.V.L and J.J.L designed the research. W.V.L conducted the research. J.J.L supervised the project. W.V.L and J.J.L. discussed the results and contributed to the manuscript writing.

## Additional information

**Competing interests:** The authors declare no competing interests.

