## [Peer Review File · Nature Communications]

Reviewers' comments:

Reviewer #3 (Remarks to the Author):

The authors of "scImpute: accurate and robust imputation for single cell RNA-seq data" propose a method to deal with an important problem of scRNA-seq dropout events. The algorithm suggests to use a statistical model that can be summarized in two main parts: identification of candidate dropout values based on Gaussian mixture model, imputation of candidate values based on the similarity principles between the cells.

While the idea of imputing missing values in the single cell RNA-seq data based on similarity principle is not novel and is already implemented in other approaches (MAGIC for example), combining with a noise model for selecting dropout values is a potentially relevant contribution.

The methodology of the paper is clear and consistent with demonstrated capabilities to enhance visualization, separation of clustering and grouping of cells, improve outcome of gene expression of differential gene expression analysis in combination with other packages.

After reading the paper carefully the following comments should be addressed:

- a) Page 2, second paragraph, the authors comment on the protocol – dependent number of dropout events. This should be addressed in more detail. It is well known that there is a negative correlation between transcriptomic complexity and dropout probability. Previously, in droplet like technologies transcriptomic complexity was low, thus translating into increased number of dropout events. New droplet technologies, like 10x, have improved the transcriptomic complexity to the level of other microfluidic technologies (plate-based or microwells). Please, provide assessment and comparisons.
- b) Figure 6) the authors demonstrate improved clustering using their methodology based on PBMC dataset. In the raw data did the authors use zeros for dropouts or they masked them as the missing values?
- c) To demonstrate the validity of their method the author use a strategy to compare their approach with other imputation methodologies, like MAGIC or SAVER. This is a good starting point, given that they are taking the same perspective, i.e. imputations. However, it is important to consider a comparison with other approaches, like those which assume zero-inflated distributions. A possible comparison would be with differentially expressed genes analysis. Do the authors know if scImpute makes the difference with MAST or PAGODA/SCDE codes?
- d) The authors claim that scImpute can be used as part of the pipeline of differential expression analysis and they do it citing works by Kharchenko et al. and Finak et al. This is a misleading (when no contradictory) statement given the fact that the works above cited are already using models of zero-inflated distributions and therefore the dropout phenomenon is already addressed. In the opinion of this referee using both approaches at the same time would most likely introduce undesired effects on the data analysis.
- e) Concerning the methodology used to identify dropouts the authors develop a method to calculate

probability of dropout for a gene in a cell. In the section devoted to it the authors comment on how the dropouts generate a bimodal distribution in the gene expression. Even if this is true, it is also known that the stochasticity of the gene expression also makes the genes to have a bimodal distribution. Is this method able to disentangle between dropout and stochastic expression? In other words, are the bimodal distribution of all the genes removed after applying scImpute algorithm?

f) In section 4.3 the authors propose a Gamma + Normal distribution to model the density function of gene expression in a subpopulation. In the literature, one can find logistic + negative binomial fitting based on empirical data, like in the works by Kharchenko et al. (the authors comment on this too). The authors should comment on why and how novel model is better than those previously used on the literature. Moreover, given that the ones based on logistic + negative binomial are coming from an empirical relation between dropout rate and mean gene expression level, why is it interesting to go beyond this model.

g) Section 4.3, the authors point that the first step is to cluster cells by using spectral clustering. The probability of dropout for a gene in a cell is then calculated based on the gene expression and variance distributions for a particular population (cluster). A potential weakness of this approach is that the clustering is not always the same i.e. different choice of parameters make to change the clusters (even if the number of them is preserved to some extent). That could in principle create different dropout probability values for the same gene in the same population depending on how the clusters are generated. It is therefore crucial to demonstrate the robustness of this way of assigning probability of being drop out against changes in parameters of clustering.

To summarize, I consider the paper to be nicely written, addressing an important problem, although the main strategy is not terribly innovative. Points above should be addressed.

Reviewer #4 (Remarks to the Author):

In general, I am not a big fan of imputation methods for scRNA-seq, because these tend to remove true biological signal. As they authors have carefully discussed in their paper, zeros can be driven by both technical and biological variation, and as such it is important to consider this before imputing zeros. I am happy to see that the authors have thought about this with their proposed mixture model-based approach. I am also pleased to see that they authors have significantly improved their paper based on comments I had provided on an earlier version.

The authors have added several datasets including the one I had recommended. They have also provided a comparison with a tool developed for scRNA-seq (MAST). So I am very happy with the paper as it stands. I have a few (rather) minor comments below:

-In the abstract, the authors say: "scImpute is shown to correct false zero counts" I don't like the term false zero. We don't know for sure that these are false, I encourage the authors to find a better term.

- The cell cycle datasets used in the paper has been shown to have problems due to improper normalization and batch effects. See:

McDavid, A., Finak, G., Gottardo, R., 2016. The contribution of cell cycle to heterogeneity in single-cell RNA-seq data. *Nat. Biotechnol.* 34, 591–593.

I am not necessarily asking the authors to remove this dataset but I would ask them to either apply their method on the data normalized for sequencing depth (not just ERCC normalization) as described in the paper above or at least discuss the limitation of this analysis. Either way the above paper

should be discussed.

- MAST citation: Given that the authors use MAST, the MAST paper should be cited:

McDavid, A., Finak, G., Gottardo, R., 2016. The contribution of cell cycle to heterogeneity in single-cell RNA-seq data. *Nat. Biotechnol.* 34, 591–593.

- Citation: I would also ask that the author cite the following paper, which is the first work on modeling dropout events in single-cell genomics data:

McDavid, A., Finak, G., Chattopadhyay, P.K., Dominguez, M., Lamoreaux, L., Ma, S.S., Roederer, M., Gottardo, R., 2013. Data exploration, quality control and testing in single-cell qPCR-based gene expression experiments. *Bioinformatics* 29, 461–467.

I hate to self advertise our work, but I do think these papers are relevant here and should be cited to properly acknowledge prior work.

- Simulated data (page 4): The authors say “they become less separated in the raw data”. The raw data are still well separated, so I was wondering if this statement could be quantified somehow.

- Figure 6: Some of the colors are difficult to distinguish, perhaps labels could be added to the figure so that differences in populations can be easily seen?

- Page 11: “...+1.01” why is 1.01 used instead of 1?

- Tuning parameters (K and t): I like that the authors have provided a sensitivity analysis. In addition to this, I think it would be good to provide recommendations on default parameters? Can these parameters be selected automatically?

Dear Editor and Reviewers,

Thank you for your helpful and detailed comments for us to improve our manuscript. We have thoroughly modified and edited our manuscript based on the reviewers' comments, and we have made our manuscript comply with editorial policies.

Below please find our responses to reviewers' comments. Please note that the paragraph and page numbers we refer to in our responses correspond to the highlighted version of our revised manuscript.

Reviewers' comments:

Reviewer #3 (Remarks to the Author):

The authors of "scImpute: accurate and robust imputation for single cell RNA-seq data" propose a method to deal with an important problem of scRNA-seq dropout events. The algorithm suggests to use a statistical model that can be summarized in two main parts: identification of candidate dropout values based on Gaussian mixture model, imputation of candidate values based on the similarity principles between the cells.

While the idea of imputing missing values in the single cell RNA-seq data based on similarity principle is not novel and is already implemented in other approaches (MAGIC for example), combining with a noise model for selecting dropout values is a potentially relevant contribution.

The methodology of the paper is clear and consistent with demonstrated capabilities to enhance visualization, separation of clustering and grouping of cells, improve outcome of gene expression of differential gene expression analysis in combination with other packages.

After reading the paper carefully the following comments should be addressed:

Response:

We thank the reviewer for taking the time to comment on our manuscript and give us the helpful suggestions. We have carefully addressed the reviewer's comments and questions. Please see our point-by-point responses below.

a) Page 2, second paragraph, the authors comment on the protocol – dependent number of dropout events. This should be addressed in more detail. It is well known that there is a negative correlation between transcriptomic complexity and dropout probability. Previously, in droplet like technologies transcriptomic complexity was low, thus translating into increased number of dropout events. New droplet technologies, like 10x, have improved the transcriptomic complexity to the level of other microfluidic technologies (plate-based or microwells). Please, provide assessment and comparisons.

Response:

We thank the reviewer for pointing out the necessity to discuss the protocol-dependent number of dropout events. We acknowledge that newer droplet technologies have an overall improved sensitivity in detecting mRNA molecules, but the droplet-based methods overall still have lower detection rates compared to microfluidic technologies. As the reviewer suggested, we have added a sentence to the main text (Paragraph 2 on Page 2). We have also added Supplementary Table S3, which shows that the PBMC dataset (generated from the 10x Genomics platform) has a higher proportion of zero values in the raw data, compared to all other datasets generated from the Fluidigm C1 platform used in our work. In addition, we refer the reviewer and readers to the current reference 11 for a power analysis on different protocols (Paragraph 2 on Page 2).

b) Figure 6) the authors demonstrate improved clustering using their methodology based on PBMC dataset. In the raw data did the authors use zeros for dropouts or they masked them as the missing values?

Response:

When clustering the cells based on the raw data, we did not mask the zero expressions as missing values. This is because (1) existing work that did not account for dropout events mostly directly performed clustering on the raw data with all zero expression values (or after dimension reduction); (2) it is not possible to distinguish missing data from true zero expression directly from the raw data (without applying scImpute). We have added this clarification to the legend of Figure 6.

c) To demonstrate the validity of their method the author use a strategy to compare their approach with other imputation methodologies, like MAGIC or SAVER. This is a good starting point, given that they are taking the same perspective, i.e. imputations. However, it is important to consider a comparison with other approaches, like those which assume zero-inflated distributions. A possible comparison would be with differentially expressed genes analysis. Do the authors know if scImpute makes the difference with MAST or PAGODA/SCDE codes?

Response:

We thank the reviewer for this suggestion. We agree with the reviewer that aside from comparing scImpute with other imputation methods like MAGIC or SAVER, it would be helpful to compare scImpute with other non-imputation methods that implicitly account for dropout events, such as those differential gene expression methods that model zero-inflated distributions. Although it is difficult to have a direct and fair comparison of imputation and differential gene expression methods, we assessed the performance of (1) **DESeq2(sc)**: DESeq2 on the raw single-cell data vs. **scImpute+DESeq2(sc)**: DESeq2 on the imputed single-cell data vs. **DESeq2(bulk)**: DESeq2 on the bulk data (zero-inflation not considered) and (2) **MAST(sc)**: MAST on the raw single-cell data vs. **scImpute+MAST(sc)**: MAST on the imputed single-cell data vs. **MAST(bulk)**: MAST on the bulk data (zero-inflation considered). We also tried to include SCDE in the analysis, but its software always aborted with errors, as other researchers have reported and discussed in online bioinformatic forums (e.g., <https://www.biostars.org/p/262934/>). Our results show that in both (1) and (2), the differentially expressed (DE) genes identified from the imputed data resemble the DE genes identified from the bulk data of the same

cell types, much more than those from the raw data did. In addition, we found that both DESeq2 and MAST, the former does not account for dropout events and the latter does, were able to detect more biologically meaningful DE genes from the imputed data, but they were not able to identify the same genes from the raw data. These results are notable in that by imputing the dropout events, scImpute also improves the performance of MAST, which accounts for dropout events by assuming zero-inflated distributions. Detailed description of this comparison and results is in Section 2.3.

d) The authors claim that scImpute can be used as part of the pipeline of differential expression analysis and they do it citing works by Kharchenko et al. and Finak et al. This is a misleading (when no contradictory) statement given the fact that the works above cited are already using models of zero-inflated distributions and therefore the dropout phenomenon is already addressed. In the opinion of this referee using both approaches at the same time would most likely introduce undesired effects on the data analysis.

Response:

Our answer to this comment is related to the reviewer's previous and next comments. In our reply to the previous comment, we showed that scImpute can improve the performance of MAST in detecting differentially expressed genes. A possible reason of why MAST also worked well for the imputed data is that even though scImpute aims to correct the dropout events, the imputed gene expression may still present a zero-inflated distribution. This may result from the stochasticity of gene expression (i.e., some genes are on-and-off among cells of the same type), as the reviewer mentioned in the next comment, or the fact that some dropouts remain uncorrected due to lack of information. We looked into the detailed methodology of SCDE (Kharchenko et al.) and MAST (Finak et al.) and found that neither of them requires a specific proportion of zero expression in their assumed zero-inflated distributions of gene expression. Therefore, these differential expression methods developed for scRNA-seq data remain applicable to the imputed data. However, we agree with the reviewer that compared with the existing methods developed for raw scRNA-seq data, new analyzing tools specifically designed for imputed scRNA-seq data may have improved performance, and we have revised the Discussion section on Page 10 accordingly.

e) Concerning the methodology used to identify dropouts the authors develop a method to calculate probability of dropout for a gene in a cell. In the section devoted to it the authors comment on how the dropouts generate a bimodal distribution in the gene expression. Even if this is true, it is also known that the stochasticity of the gene expression also makes the genes to have a bimodal distribution. Is this method able to disentangle between dropout and stochastic expression? In other words, are the bimodal distribution of all the genes removed after applying scImpute algorithm?

Response:

We thank the reviewer for commenting on the bimodal assumption of gene expression. Our method scImpute tries to distinguish dropouts from true zero expression values in two aspects. First, scImpute estimates the dropout probability of each gene in each cell, and it only imputes the expression of a gene if its dropout probability is higher than a

threshold, which is a tuning parameter with a default value 0.5, in a cell. In other words, a likely true zero expression value with a low dropout probability would not be altered by scImpute. Second, scImpute predicts the expression of a dropout in a cell based on the expression of the same gene in other similar cells. Therefore, the imputed value is non-zero only when the gene is expressed in similar cells. These two aspects also show that scImpute takes a relatively conservative approach to impute dropouts and avoid introducing new biases. To illustrate that scImpute does not remove the bimodal distribution of all the genes, we have added Figure S25 to illustrate the distributions of expression values of four example genes. This new figure shows that scImpute did not remove the stochasticity of gene expression. We have also incorporated the above information into Section 4.4 on Page 14 in the revised manuscript.

f) In section 4.3 the authors propose a Gamma + Normal distribution to model the density function of gene expression in a subpopulation. In the literature, one can find logistic + negative binomial fitting based on empirical data, like in the works by Kharchenko et al. (the authors comment on this too). The authors should comment on why and how novel model is better than those previously used on the literature. Moreover, given that the ones based on logistic + negative binomial are coming from an empirical relation between dropout rate and mean gene expression level, why is it interesting to go beyond this model.

Response:

We thank the reviewer for this comment. There are two reasons for our proposal of using a Gamma + Normal mixture model. First, we chose to apply the logarithmic transformation to the raw read counts to prevent a few large observations from being extremely influential, and the transformed values become continuous, allowing for greater flexibility for the modeling. Therefore, the mixture model of Poisson and negative binomial distribution (like the model used in Kharchenko et al.) is no longer proper to describe the transformed data, and we found a Gamma + Normal mixture as an appropriate choice that fits the transformed data well. Second, we have noticed that some work used the logistic regression (including Kharchenko et al.) to model the relationship between dropout rates and mean gene expression level. However, the logistic regression itself is only an approximation and depends on the estimation of mean gene expression. Given these reasons, we think it is better to consider dropout rate as a free parameter in the Gamma + Normal mixture model and directly estimate it based on the observed data. Compared to the logistic + negative binomial model, our model has greater flexibility and less bias, and the model performance is demonstrated by our simulation and real data results. We have incorporated the above comments and explanation into the Methods Sections 4.1 and 4.3.

g) Section 4.3, the authors point that the first step is to cluster cells by using spectral clustering. The probability of dropout for a gene in a cell is then calculated based on the gene expression and variance distributions for a particular population (cluster). A potential weakness of this approach is that the clustering is not always the same i.e. different choice of parameters make to change the clusters (even if the number of them is preserved to some extent). That could in principle create different dropout probability values for the same gene in the same population depending on how the clusters are generated. It is therefore crucial to demonstrate the

robustness of this way of assigning probability of being drop out against changes in parameters of clustering.

Response:

We thank the reviewer for raising this important point. We agree that it is necessary to demonstrate the robustness of scImpute. Therefore, we have illustrated the distribution of the estimated dropout probabilities and the robustness of scImpute to the choice of cluster numbers in Figure S24, which shows that scImpute is robust to different tuning parameter choices. In the Methods section, we have also clarified that after the clustering step, scImpute uses a non-negative linear regression to further select similar cells, so the imputation process is not fully dependent on the clustering results. In addition, since the similarity among individual cells is a relative but not an absolute concept, the choice of cluster numbers actually gives users the freedom to impute the data based on their desired level of resolution. If a user is only interested in the differences among the major clusters, a relatively small cluster number could be used, and scImpute can borrow more information between cells; otherwise, the user could select a relatively large cluster number, and scImpute would be more conservative in its imputation. We have added the above information into the Discussion section (the last paragraph on Page 10 and the first paragraph on Page 11).

To summarize, I consider the paper to be nicely written, addressing an important problem, although the main strategy is not terribly innovative. Points above should be addressed.

Reviewer #4 (Remarks to the Author):

In general, I am not a big fan of imputation methods for scRNA-seq, because these tend to remove true biological signal. As they authors have carefully discussed in their paper, zeros can be driven by both technical and biological variation, and as such it is important to consider this before imputing zeros. I am happy to see that the authors have thought about this with their proposed mixture model-based approach. I am also pleased to see that they authors have significantly improved their paper based on comments I had provided on an earlier version.

The authors have added several datasets including the one I had recommended. They have also provided a comparison with a tool developed for scRNA-seq (MAST). So I am very happy with the paper as it stands. I have a few (rather) minor comments below:

Response:

We appreciate that the reviewer takes his/her time to review our revised manuscript. We are also thankful that the reviewer's comments have greatly helped us improve our method and manuscript. Please find our detailed responses as listed below.

-In the abstract, the authors say: "scImpute is shown to correct false zero counts" I don't like the term false zero. We don't know for sure that these are false, I encourage the authors to find a better term.

Response:

We thank the reviewer for this advice. We have changed the sentence "scImpute is shown to correct false zero counts" to "scImpute is shown to identify likely dropout events".

- The cell cycle datasets used in the paper has been shown to have problems due to improper normalization and batch effects. See:

McDavid, A., Finak, G., Gottardo, R., 2016. The contribution of cell cycle to heterogeneity in single-cell RNA-seq data. Nat. Biotechnol. 34, 591–593.

I am not necessarily asking the authors to remove this dataset but I would ask them to either apply their method on the data normalized for sequencing depth (not just ERCC normalization) as described in the paper above or at least discuss the limitation of this analysis. Either way the above paper should be discussed.

Response:

We thank the reviewer for pointing out the issue with the dataset. As we discussed in the Methods section, we did normalize the data for sequencing depth, but we did not emphasize this step in our previous manuscript. As the reviewer suggested, we have added sentences to emphasize the normalization step and added the above reference to the revised manuscript so that the readers can understand this dataset and our results better. The reviewer can refer to Section 4.1 on Page 11.

- MAST citation: Given that the authors use MAST, the MAST paper should be cited:

McDavid, A., Finak, G., Gottardo, R., 2016. The contribution of cell cycle to heterogeneity in single-cell RNA-seq data. *Nat. Biotechnol.* 34, 591–593.

- Citation: I would also ask that the author cite the following paper, which is the first work on modeling dropout events in single-cell genomics data:

McDavid, A., Finak, G., Chattopadhyay, P.K., Dominguez, M., Lamoreaux, L., Ma, S.S., Roederer, M., Gottardo, R., 2013. Data exploration, quality control and testing in single-cell qPCR-based gene expression experiments. *Bioinformatics* 29, 461–467.

I hate to self advertise our work, but I do think these papers are relevant here and should be cited to properly acknowledge prior work.

Response:

We thank the reviewer for pointing us to the above references relevant to our work. We have cited them to acknowledge their important contribution and provide readers with more comprehensive background information.

- *Simulated data (page 4): The authors say “they become less separated in the raw data”. The raw data are still well separated, so I was wondering if this statement could be quantified somehow.*

Response:

We thank the reviewer for this suggestion. We have calculated the within-cluster sum-of-squares of the complete, raw, and imputed data and used this measure to quantify the separation of different cell types. The within-cluster sum-of-squares of the complete, raw, and imputed data by scImpute, MAGIC, and SAVER are 94, 2646, 152, 1706, and 1826, respectively. The changes are reflected in Figure 4 and also in the main text (the last paragraph on Page 5).

- *Figure 6: Some of the colors are difficult to distinguish, perhaps labels could be added to the figure so that differences in populations can be easily seen?*

Response:

We agree with the reviewer and have updated Figure 6 to show cell type labels next to main clusters.

- *Page 11: “...+1.01” why is 1.01 used instead of 1?*

Response:

Because in case of zero expression (i.e., $\log(\text{count} + 1)$ would give a zero value when $\text{count} = 0$), there would be infinite values in the parameter estimation of gamma distribution in the mixture model. As a consequence, we have to add a small value 0.01 to 1.

- *Tuning parameters (K and t): I like that the authors have provided a sensitivity analysis. In*

*addition to this, I think it would be good to provide recommendations on default parameters?
Can these parameters be selected automatically?*

Response:

We are thankful that the reviewer found the sensitivity analysis helpful. As we have demonstrated in Figure S24c, a parameter t between 0.25 and 0.75 only have a minute effect on the identification of dropout events, so we recommend a default value of 0.5, as set in the `scImpute` R package. As for the parameter K , we suggest that users select its value based on the initial clustering results of raw data, and we did not provide a default value, because the number of cell subpopulations may vary greatly from dataset to dataset. In the Methods section, we have introduced that after the clustering step, `scImpute` uses the non-negative linear regression to further select similar cells, so the imputation process is not fully dependent on the clustering results. In addition, since the similarity among individual cells is a relative and not an absolute concept, the choice of cluster numbers gives the users the freedom to impute the data based on their desired level of resolution. If the users are only interested in the differences among the major clusters, then a relatively small K could be used, and `scImpute` can borrow more information between cells; otherwise, the users can select a relatively large K and `scImpute` would be more conservative. We have revised the Discussion section to reflect the above information.

REVIEWERS' COMMENTS:

Reviewer #3 (Remarks to the Author):

The authors correctly addressed potential concerns.

Please note that while Reviewer 4 doesn't have comments to the author, in his/her comments to the editor, he/she thinks the authors have properly addressed the comments.

REVIEWERS' COMMENTS:

Reviewer #3 (Remarks to the Author):

The authors correctly addressed potential concerns.

Please note that while Reviewer 4 doesn't have comments to the author, in his/her comments to the editor, he/she thinks the authors have properly addressed the comments.

Response:

We thank both reviewers for taking the time to review our re-submission and for their insightful suggestions on our manuscript.